**Data Availability Statement:** All relevant data are within the paper and its Supporting Information files. Our ethical approval from the Swedish Ethical Review Authority (EPA; No. 2020-02231) limits us

# Physiological respiratory parameters in pre-hospital patients with suspected COVID-19: A prospective cohort study

**Johan Mälberg** [1,2,3] *, **Nermin Hadziosmanovic** [4], **David Smekal** [1,2,3]

**1** Department of Surgical Sciences-Anesthesia and Intensive Care, Uppsala University, Uppsala, Sweden, **2** Uppsala Center for Prehospital Research, Uppsala University Hospital, Uppsala, Sweden, **3** Uppsala Emergency Medical Service, Uppsala University Hospital, Uppsala, Sweden, **4** Uppsala Clinical Research Center, Uppsala University, Uppsala, Sweden

* johan.malberg@surgsci.uu.se

## Abstract

### Background

The COVID-19 pandemic has presented emergency medical services (EMS) worldwide with the difficult task of identifying patients with COVID-19 and predicting the severity of their illness. The aim of this study was to investigate whether physiological respiratory parameters in pre-hospital patients with COVID-19 differed from those without COVID-19 and if they could be used to aid EMS personnel in the prediction of illness severity.

### Methods

Patients with suspected COVID-19 were included by EMS personnel in Uppsala, Sweden. A portable respiratory monitor based on pneumotachography was used to sample the included patient's physiological respiratory parameters. A questionnaire with information about present symptoms and background data was completed. COVID-19 diagnoses and hospital admissions were gathered from the electronic medical record system. The physiological respiratory parameters of patients with and without COVID-19 were then analyzed using descriptive statistical analysis and logistic regression.

### Results

Between May 2020 and January 2021, 95 patients were included, and their physiological respiratory parameters analyzed. Of these patients, 53 had COVID-19. Using adjusted logistic regression, the odds of having COVID-19 increased with respiratory rate (95% CI 1.000–1.118), tidal volume (95% CI 0.996–0.999) and negative inspiratory pressure (95% CI 1.017–1.152). Patients admitted to hospital had higher respiratory rates (p<0.001) and lower tidal volume (p = 0.010) compared to the patients who were not admitted. Using adjusted logistic regression, the odds of hospital admission increased with respiratory rate (95% CI 1.081–1.324), rapid shallow breathing index (95% CI 1.006–1.040) and dead space percentage of tidal volume (95% CI 1.027–1.159).

to only sharing anonymized data that cannot be used to identify individual participants. As the data set used for the statistical analysis in this study contains sensitive information that could be used to identify individual participants, and could therefore compromise their privacy, we are unable to share this data in full. We have after consulting the Swedish Ethical Review Authority, anonymized the participants by removing age, gender, inclusion time and date from the minimal data set that we include as supporting information with this manuscript. This was done to in order to comply with our ethical approval, Swedish data protection laws and the EU General Data Protection Regulation in order to ensure the participants privacy. The statistical analyses themselves are include in full as supporting information as they contain no sensitive information. Researchers who meet the criteria for access to confidential information and are interested in the full data set are kindly asked to contact sten.rubertsson@surgsci.uu.se who represents the institution of surgical sciences at Uppsala University which owns the data that this study is based on.

**Funding:** The authors received no specific funding for this work.

**Competing interests:** The authors have declared that no competing interests exist.

## Conclusion

Patients taking smaller, faster breaths with less pressure had higher odds of having COVID-19 in this study. Smaller, faster breaths and higher dead space percentage also increased the odds of hospital admission. Physiological respiratory parameters could be a useful tool in detecting COVID-19 and predicting hospital admissions, although more research is needed.

## Introduction

Since the emergence of severe acute respiratory syndrome coronavirus 2 (SARS-CoV-2) at the end of 2019, the virus has spread rapidly throughout the world causing the corona virus disease 2019 (COVID-19) pandemic, resulting in major disruptions to everyday life and a large number of deaths. It has also placed a great burden on the healthcare sector with high numbers of hospital and intensive care unit (ICU) admissions [1].

Most Patients with COVID-19 have mild symptoms such as fever, cough, fatigue and dyspnea usually appearing within approximately 5 days [2]. For some however, the disease progresses to a more severe state with one or more complications and where ICU-care might be needed. The most common serious complication is acute respiratory distress syndrome (ARDS) [3].

In the pre-hospital setting, the pandemic presents numerous challenges for the emergency medical services (EMS) responding to patients with suspected COVID-19. One of these challenges is identifying patients with an actual COVID-19 infection [4,5]. Even more challenging is the prediction of the severity of the disease and future need of healthcare and ICU admissions.

Physiological respiratory parameters has been postulated as a way to predict severity and ICU admissions in other diseases of the lungs, such as chronic obstructive pulmonary disease (COPD) and acute respiratory failure [6,7] but it remains unclear if this is the case in patients with COVID-19. It has been suggested that dyspnea could be used as a predictor for ICU admission in this patient category but further research is needed [8]. One theoretical disadvantage of that predictor is the presence of case reports showing that COVID-19 patients can have silent hypoxemia, meaning hypoxemia without dyspnea, which could make detection of hypoxic COVID-19 patients more difficult [9,10]. Silent hypoxemia may also be present in COVID-19 patients during pulse elevating activities such as brisk walking, which is one method that has been used to discover dyspnea and hypoxemia in these patients [11]. Other means of discovering hypoxic COVID-19 patients might thus be necessary.

The aim of this study was therefore to measure physiological respiratory parameters and other vital signs in patients with suspected COVID-19 in the pre-hospital setting and identify any correlations between these findings and the occurrence of positive COVID-19 polymerase chain reaction (PCR) tests, morbidity, and mortality among these patients.

## Materials and methods

### Study design and participants

This prospective, cohort study was conducted in the Region of Uppsala, Sweden. The study was approved by the Swedish National Ethical Review Authority (EPA; No. 2020–02231). Informed written consent was obtained from the patients. The Declaration of Helsinki and its

subsequent revisions were adhered to. All patients receiving an ambulance between May 2020 and January 2021 aged 18 years or older, with confirmed or suspected COVID-19 were eligible for inclusion by EMS personnel. The sole exclusion criterion was the inability to give informed consent (i.e. severe respiratory distress/failure). The suspicion of COVID-19 was based on patients presenting one or more of the following symptoms: Fever, cough, dyspnea, runny nose, or sore throat. The included patients were selected by the EMS personnel.

## Procedure

A portable respiratory monitor based on pneumotachography [12] manufactured by MBMed called FluxMed GrH ℝ was used for the sampling of physiological respiratory parameters. The sampling procedure was initiated by the EMS personnel asking the patients to breathe normally in a mouthpiece connected via a flexible tube connected to the sensor device for approximately one minute. During this time, every individual breath was sampled and data was stored on a portable computer connected to the FluxMed GrH ℝ device.

Data sampled from the FluxMed GrH ℝ device were: Respiratory rate (RR), minute ventilation (MV), rapid shallow breathing index (RSBI), inspiratory tidal volume (Vti), peak inspiratory flow (PIF), peak expiratory flow (PEF), peak negative inspiratory pressure (NIP), peak expiratory pressure (PEP), end tidal $CO_2$ (Etco2) and physiological dead space volume (Vd).

The device was already in use before the pandemic during cardiopulmonary resuscitation and all personnel were trained in the use of the device prior to this study. Nevertheless, all EMS personnel were trained in handling the FluxMed GrH ℝ device and the portable computer in this new setting and on this new group of patients before the start of the study.

After the sampling procedure, a questionnaire with questions regarding the presence of symptoms including fever, cough, dyspnea, sore throat, or runny nose were completed. Length and weight were also recorded for every patient. The patients gender and weight was later used to calculate predicted body weight (PBW) according to NHLBI ARDS Network [13]. PBW was then used to calculate ideal Vti, where 7ml/kg was used [14]. The EMS personnel would then measure the participants' respiratory rate, body temperature, pulse, and blood oxygen saturation and all these parameters were noted on that same questionnaire.

The included patients were monitored via the electronic medical record system (EMRS) after inclusion and information regarding COVID-19 diagnosis was collected. COVID-19 diagnosis was defined as a positive PCR test in connection to the episode of illness when the inclusion was made. No COVID-19 diagnosis was defined as negative PCR test or negative clinical diagnosis in connection to the episode of illness when the patient was included. Additional information gathered from the EMRS was treatment with dexamethasone or remdesivir, if the patient had received supplementary oxygen, oxygen flow rate, non-invasive ventilation (NIV) or if the patient was intubated and received ventilator care. Demographic data and pre-existing comorbidities were also obtained from the EMRS.

## Outcomes

The primary outcome was if there were any differences in the physiological respiratory parameters between patients with COVID-19 and patients without COVID-19.

Secondary outcomes were if there were any correlations between physiological respiratory parameters and hospital admissions, length of stay in hospital and treatments given.

## Statistical analysis

The characteristics of the study population were summarized by COVID-19 diagnosis, with categorical variables presented as counts and proportions, and continuous variables as

medians and interquartile ranges. To investigate differences across the groups of patients, categorical variables were compared with the Fishers exact test. Continuous variables were compared using nonparametric Mann-Whitney U-tests.

The associations between physiological respiratory parameters and COVID-19 was assessed using logistic regression models. In addition to the univariate models, the relationship between physiological respiratory parameters and COVID-19 was also analyzed in multivariate models adding age, sex, body mass index (BMI), and risk factors as covariates. Age and BMI was analyzed as continuous variables. Risk factors were analyzed as categorical variables.

Results are presented including the number of observations, odds ratios (OR) with 95% confidence intervals (CI), and p-values. All analyses were performed using IBM SPSS Statistics 27 software. A 2-sided p value of <0.05 was considered statistically significant.

## Results

Between May 23, 2020 and January 22, 2021, a total of 116 patients were selected by the EMS personnel to participate in the study and underwent the sampling procedure. Sixteen of the included patients had respiratory data that were of very poor quality due to technical issues and were therefore excluded. One patient was included twice, and the second inclusion of this patient was excluded. Four of the patients had no PCR-test or clinical diagnosis and were excluded. Of the 95 patients included in the final analysis, 53 had COVID-19. The inclusion chart can be seen in Fig 1.

The characteristics of the included patients can be seen in Table 1. When comparing COVID-19 positive with COVID-19 negative patients, the COVID-19 positive group were significantly younger (50 years old vs. 71 years old, p = 0.021) and had a higher weight (83 kg vs. 70 kg, p = 0.001) and BMI (28.6 vs. 25.1, p = 0.009). They also had a lower prevalence of heart failure (7.5% vs. 26.2%, p = 0.022) and coronary heart disease (5.7% vs. 26.2%, p = 0.008). In the positive group, occurrences of fever (51.9% vs 26.2%, p = 0.019) and cough (86.8% vs. 64.3%, p = 0.014) were higher. Pulse (104.0 beats per minute vs. 92.0 beats per minute, p = 0.017) and body temperature (38.0 C˚ vs. 37.25 C˚, p = 0.002) were also higher in this group. Eight of the included patients died within 30 days of inclusion. Four of these deaths were due to COVID-19. Hospital admission rates were 67.9% in the positive group and 54.8% in the negative group.

Treatments and time in hospital can be seen in Table 2. Oxygen treatment was more prevalent (80.6% vs. 43.5%, p = 0.005) in the COVID-19 positive group. There were no differences in hospital length of stay between the groups.

There were no differences in the physiological respiratory parameters between the COVID-19 negative and positive groups (Table 3).

Adjusting for age, sex, BMI and risk factors in a logistic regression model, the odds of having COVID-19 increased with increasing RR (OR 1.057, 95% CI 1.000–1.118, p = 0.049), RSBI (OR 1.018, 95% CI 1.004–1.032, p = 0.014) and NIP (OR 2.212, 95% CI 1.184–4.132, p = 0.013). The odds of having COVID-19 also increased with decreasing Vti (OR 0.998, 95% CI 0.996–0.999, p = 0.001), Vti percentage of ideal Vti (OR 0.990, 95% CI 0.984–0.996, p = 0.001) and PIF (OR 0.970, 95% CI 0.947–0.994, p = 0.015) (Table 4). The complete logistic regression models can be found in S4 Dataset.

The patients who were admitted to hospital had higher RR (p<0.001), RSBI (p<0.001), Vd percentage of Vti (p<0.001), lower Vti (p = 0.010) and Vti percentage of ideal Vti (p = 0.001) than the patients who were not admitted. This was regardless of having COVID-19 or not. The physiological respiratory parameters, hospital admissions and COVID-19 are described in

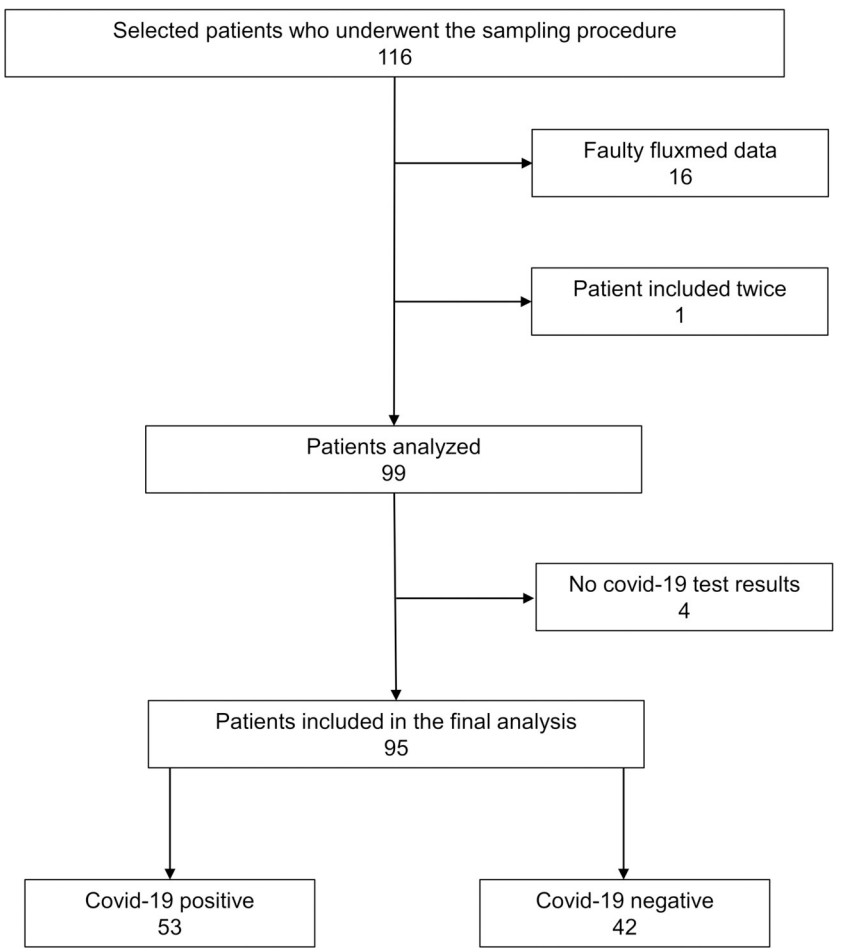

**Fig 1. Inclusion chart.**

more detail in Table 5. Significant physiological respiratory parameters are explored further in Fig 2. Full Kruskal-Wallis test data can be found in S5 Dataset.

Adjusting for age, sex, BMI, risk factor categorical and COVID-19, the odds of hospital admission increased with increasing RR (OR 1.196, 95% CI 1.081–1.324, p = 0.001), RSBI (OR 1.023, 95% CI 1.006–1.040, p = 0.009) and Vd percentage of Vti (OR 1.091, 95% CI 1.027–1.159, p = 0.005) (Table 6). The complete logistic regression models can be found in S6 Dataset.

## Discussion

In this study, the odds of having COVID-19 increased with higher respiratory rate (OR 1.057, 95% CI 1.000–1.118, p = 0.049), higher rapid shallow breathing index (OR 1.018, 95% CI 1.004–1.032, p = 0.014) and higher negative inspiratory pressure (OR 2.212, 95% CI 1.184–4.132, p = 0.013). The odds also increased with lower tidal volume (OR 0.998, 95% CI 0.996–0.999, p = 0.001), tidal volume percentage of ideal tidal volume (OR 0.990, 95% CI 0.984–0.996, p = 0.001) and peak inspiratory flow (OR 0.970, 95% CI 0.947–0.994, p = 0.015). This indicates that taking smaller, faster breaths with lower flow and pressure is associated with higher odds of having COVID-19. Few if any studies have explored the physiological respiratory parameters of COVID-19 patients prior to hospital admission. There are however some

**Table 1. Patient characteristics.**

| | COVID-19 diagnosis | | |
| --- | --- | --- | --- |
| | COVID-19 negative (n = 42) | COVID-19 positive (n = 53) | p-value[a] |
| **Demographical data:** | | | |
| Male sex, n (%) | 19 (45.2) | 29 (54.7) | 0.412 |
| Age, median (IQR) | 71.5 (42.0) | 50.0 (15.0) | 0.021* |
| Length (cm), median (IQR) | 170.0 (17.0) | 173.0 (15.0) | 0.282 |
| Weight (kg), median (IQR) | 70.0 (18.0) | 83.0 (22.3) | 0.001* |
| BMI, median (IQR) | 25.1 (6.7) | 28.6 (5.9) | 0.009* |
| **Risk factors:** | | | |
| Hypertension, n (%) | 22 (52.4) | 22 (41.5) | 0.309 |
| Heart failure, n (%) | 11 (26.2) | 4 (7.5) | 0.022* |
| Diabetes, n (%) | 7 (16.7) | 11 (20.8) | 0.793 |
| Coronary heart disease, n (%) | 11 (26.2) | 3 (5.7) | 0.008* |
| COPD, n (%) | 3 (7.1) | 1 (1.9) | 0.318 |
| Cancer, n (%) | 7 (16.7) | 4 (7.5) | 0.206 |
| Kidney disease, n (%) | 6 (14.3) | 2 (3.8) | 0.133 |
| Liver disease, n (%) | 1 (2.4) | 0 (0.0) | 0.442 |
| Cerebrovascular disease, n (%) | 6 (14.3) | 4 (7.5) | 0.329 |
| **Risk factors categorical** | | | 0.060[b] |
| 0 risk factors | 14 (33.3) | 29 (54.7) | |
| 1–2 risk factors | 13 (31.0) | 15 (28.3) | |
| ≥3 risk factors | 15 (35.7) | 9 (17.0) | |
| **Presenting symptoms:** | | | |
| Fever, n (%) | 11 (26.2) | 27 (51.9) | 0.019* |
| Cough, n (%) | 27 (64.3) | 46 (86.8) | 0.014* |
| Dyspnea, n (%) | 31 (73.8) | 43 (81.1) | 0.459 |
| Runny nose, n (%) | 11 (26.2) | 23 (43.4) | 0.090 |
| Sore throat, n (%) | 12 (28.6) | 20 (38.5) | 0.384 |
| **Vital parameters:** | | | |
| Respiratory rate (breaths per minute), median (IQR) | 20.0 (8.0) | 22.0 (12.0) | 0.295 |
| Body temperature (C˚), median (IQR) | 37.4 (1.1) | 38.0 (1.6) | 0.002* |
| Oxygen saturation (%), median (IQR) | 95.5 (8.0) | 96.0 (6.0) | 0.842 |
| Pulse (beats per minute), median (IQR) | 92.0 (19.0) | 104.0 (28.0) | 0.017* |
| **Subsequent care:** | | | |
| Admitted to hospital, n (%) | 23 (54.8) | 36 (67.9) | 0.208 |
| **Mortality** | | | |
| Deceased within 30 days after inclusion, n (%) | 3 (7.1) | 5 (9.4) | 1.000 |
| Deceased within 60 days after inclusion, n (%) | 0 (0) | 0 (0) | NA |

Abbreviations: IQR: Interquartile range; BMI: Body mass index; COPD: Chronic obstructive pulmonary disease.

[a] P-values calculated using Fishers exact test in dichotomous variables and Mann-Whitney-U test in continuous variables.

[b] P-value value calculated using Fisher-Freeman-Halton exact test.

* P-value of ≤ 0.05.

studies that have investigated lung volumes and physiological respiratory parameters in COVID-19 patients during and after hospitalization. One study of CT-images of COVID-19 patients found that they had significantly reduced lung volumes [15]. Another study found that 4 months after the illness, many of the patients, especially those with severe COVID-19,

**Table 2. Time and treatments in hospital.**

| | COVID-19 diagnosis | | |
|---|---|---|---|
| **Admitted to hospital:** | **COVID-19 negative (n = 23)** | **COVID-19 positive(n = 36)** | **P-value[a]** |
| Hospital length of stay (h), median (IQR) | 149.0 (128.0) | 148.0 (193.3) | 0.834 |
| Oxygen treatment, n (%) | 10 (43.5) | 29 (80.6) | 0.005* |
| Optiflow, n (% of oxygen treatment) | 0 (0.0) | 12 (41.4) | 0.017* |
| Non-invasive ventilation, n (% of oxygen treatment) | 1 (10.0) | 5 (17.2) | 1.000 |
| Dexamethasone, n (%) | 0 (0.0) | 16 (44.4) | 0.000* |
| Remdesivir, n (%) | 0 (0.0) | 6 (16.7) | 0.072 |
| Admitted to ICU, n (%) | 1 (4.3) | 7 (19.4) | 0.133 |
| ICU length of stay (h), median (IQR) | 398.0 | 127.0 (215.0) | NA |
| Ventilator care, n (% of admitted to ICU) | 1 (100%) | 3 (42.9%) | 1.000 |
| Time on ventilator (h), median (IQR) | 283.0 | 266.0 | NA |

Abbreviations: IQR: Interquartile range; ICU: Intensive care unit.

[a] P-values calculated using Fishers exact test in dichotomous variables and Mann-Whitney-U test in continuous variables.

* P-value of $\leq$ 0.05.

had reduced lung volumes and affected physiological respiratory parameters [16]. Our results might indicate that even in the early stages of the disease, before hospitalization, patients with COVID-19 encountered by the EMS could have detectable changes in their physiological respiratory parameters.

In our cohort, patients with COVID-19 had higher BMI than those without. These results are well in line with previous studies that have found high BMI to be a risk factor for severe COVID-19 [17]. Patients with COVID-19 were 21.5 years younger than patients without COVID-19 in our study, which is surprising seeing as age has been proven to be a strong risk factor for severe COVID-19 [18]. However, in Sweden as of week 16 2021, the median age of

**Table 3. Physiological respiratory parameters and COVID-19.**

| | COVID-19 diagnosis | | |
|---|---|---|---|
| **Physiological respiratory parameters, medians (IQR)** | **COVID-19 negative (n = 42)** | **COVID-19 positive (n = 53)** | **p-value[a]** |
| RR (breaths/min) | 20.0 (12.5) | 20.0(14.5) | 0.702 |
| MV (L/min) | 12.2 (9.0) | 11.6 (9.0) | 0.805 |
| RSBI (breaths/min/L) | 34.1 (53.7) | 34.8 (54.4) | 0.454 |
| Vti (ml) | 563.0 (929.6) | 568.0 (580.0) | 0.519 |
| Vti percentage of ideal Vti (%) | 152.1 (200.1) | 125.9 (140.4) | 0.309 |
| PIF (L/min) | 45.9 (29.8) | 42.8 (23.2) | 0.263 |
| PEF (L/min) | 42.1 (37.8) | 46.2 (29.5) | 0.517 |
| NIP (cmH2O) | -0.9. (0.9) | - 0.7 (0.7) | 0.077 |
| PEP (cmH2O) | 0.5 (1.0) | 0.6 (0.6) | 0.490 |
| EtCO2 (kPa) | 3.3 (1.2) | 3.5 (1.3) | 0.831 |
| Vd (ml) | 202.2 (93.6) | 182.9 (126.7) | 0.472 |
| Vd percentage of Vti (%) | 32.3 (19.2) | 32.8 (20.0) | 0.635 |

Abbreviations: IQR: Interquartile range; RR: Respiratory rate; MV: Minute ventilation; RSBI: Rapid shallow breathing index; Vti: Inspiratory tidal volume; PIF: Peak inspiratory flow; PEF: Peak expiratory flow; NIP: Negative inspiratory pressure; PEP: Peak expiratory pressure; EtCO2: End tidal carbon dioxide; Vd: Dead space volume.

[a] P-values calculated using Mann-Whitney-U test.

**Table 4. Logistic regression models of physiological respiratory parameters effect the odds of having COVID-19.**

| | Unadjusted model | | Adjusted model[a] | |
|---|---|---|---|---|
| Patient characteristics | OR (95% CI) | p-value | OR (95% CI) | p-value |
| Age | 0.972 (0.951–0.994) | 0.011* | | |
| Sex | 1.463 (0.648–3.299) | 0.360 | | |
| BMI | 1.076 (1.002–1.155) | 0.045* | | |
| Risk factors categorical | 0.540 (0.322–0.904) | 0.019* | | |
| **Physiological respiratory parameters** | | | | |
| RR (breaths/min) | 1.026 (0.981–1.072) | 0.262 | 1.057 (1.000–1.118) | 0.049* |
| MV (L/min) | 0.996 (0.941–1.054) | 0.887 | 0.945 (0.883–1.012) | 0.105 |
| RSBI (breaths/min/L) | 1.006 (0.996–1.016) | 0.245 | 1.018 (1.004–1.032) | 0.014* |
| Vti (ml) | 0.999 (0.999–1.000) | 0.152 | 0.998 (0.996–0.999) | 0.001* |
| Vti percentage of ideal Vti (%) | 0.996 (0.992–1.000) | 0.073 | 0.990 (0.984–0.996) | 0.001* |
| PIF (L/min) | 0.989 (0.970–1.009) | 0.287 | 0.970 (0.947–0.994) | 0.015* |
| PEF (L/min) | 1.005 (0.988–1.021) | 0.577 | 0.991 (0.972–1.011) | 0.372 |
| NIP (cmH2O) | 1.431 (0.843–2.428) | 0.185 | 2.212 (1.184–4.132) | 0.013* |
| PEP (cmH2O) | 1.208 (0.668–2.184) | 0.533 | 0.773 (0.395–1.515) | 0.453 |
| EtCO2 (kPa) | 0.946 (0.626–1.430) | 0.793 | 0.844 (0.529–1.348) | 0.477 |
| Vd (ml) | 0.999 (0.994–1.004) | 0.657 | 0.996 (0.991–1.001) | 0.130 |
| Vd percentage of Vti (%) | 1.006 (0.977–1.037) | 0.676 | 1.033 (0.993–1.075) | 0.107 |

Abbreviations: IQR: Interquartile range; OR: Odds ratio; CI: Confidence interval; BMI: Body mass index; RR: Respiratory rate; MV: Minute ventilation; RSBI: Rapid shallow breathing index; Vti: Inspiratory tidal volume; PIF: Peak inspiratory flow; PEF: Peak expiratory flow; NIP: Negative inspiratory pressure; PEP: Peak expiratory pressure; EtCO2: End tidal carbon dioxide; Vd: Dead space volume.

[a] Adjusted for age, sex, BMI, and risk factors categorical.

* P-value of ≤ 0.05.

**Table 5. Physiological respiratory parameters, hospital admissions and COVID-19.**

| | Not admitted to hospital | | Admitted to hospital | | |
|---|---|---|---|---|---|
| Physiological respiratory parameters, medians (IQR) | COVID-19 negative (n = 19) | COVID-19 positive (n = 17) | COVID-19 negative (n = 23) | COVID-19 positive (n = 36) | p-value |
| RR (breaths/min) | 16.0 (13.0) | 14.0 (9.0) | 22.0 (8.5) | 25.0 (16.5) | <0.001* |
| MV (L/min) | 14.7 (16.3) | 10.0 (9.8) | 10.3 (5.6) | 12.4 (8.9) | 0.353 |
| RSBI (breaths/min/L) | 12.6 (37.7) | 20.8 (15.5) | 45.6 (47.7) | 47.9 (55.3) | <0.001* |
| Vti (ml) | 1097.0 (1189.0) | 826.0 (508.8) | 465.0 (381.0) | 495.5 (357.8) | 0.010* |
| Vti percentage of ideal Vti | 282.8 (229.6) | 178.6 (132.6) | 98.6 (84.8) | 100.2 (91.1) | 0.001* |
| PIF (L/min) | 50.6 (33.4) | 44.7 (24.3) | 42.1 (14.5) | 41.8 (17.3) | 0.453 |
| PEF (L/min) | 45.9 (48.8) | 44.3 (31.5) | 38.5 (30.1) | 47.9 (29.9) | 0.907 |
| NIP (cmH2O) | -0.9 (1.1) | -0.7 (0.7) | -0.7 (0.5) | -0.6 (0.7) | 0.154 |
| PEP (cmH2O) | 0.7 (1.0) | 0.6 (0.9) | 0.5 (0.7) | 0.6 (0.5) | 0.683 |
| EtCO2 (kPa) | 3.5 (1.0) | 3.8 (1.0) | 3.0 (1.3) | 3.3 (1.4) | 0.248 |
| Vd (ml) | 226.5 (134.3) | 176.3 (150.2) | 186.3 (66.0) | 187.2 (131.3) | 0.537 |
| Vd percentage of Vti (%) | 22.8 (23.6) | 21.3 (11.0) | 36.0 (19.2) | 37.4 (14.8) | <0.001* |

Abbreviations: IQR: Interquartile range; RR: Respiratory rate; MV: Minute ventilation; RSBI: Rapid shallow breathing index; Vti: Inspiratory tidal volume; PIF: Peak inspiratory flow; PEF: Peak expiratory flow; NIP: Negative inspiratory pressure; PEP: Peak expiratory pressure; EtCO2: End tidal carbon dioxide; Vd: Dead space volume.

[a] P-values calculated using Kruskal-Wallis test.

* P-value of ≤ 0.05.

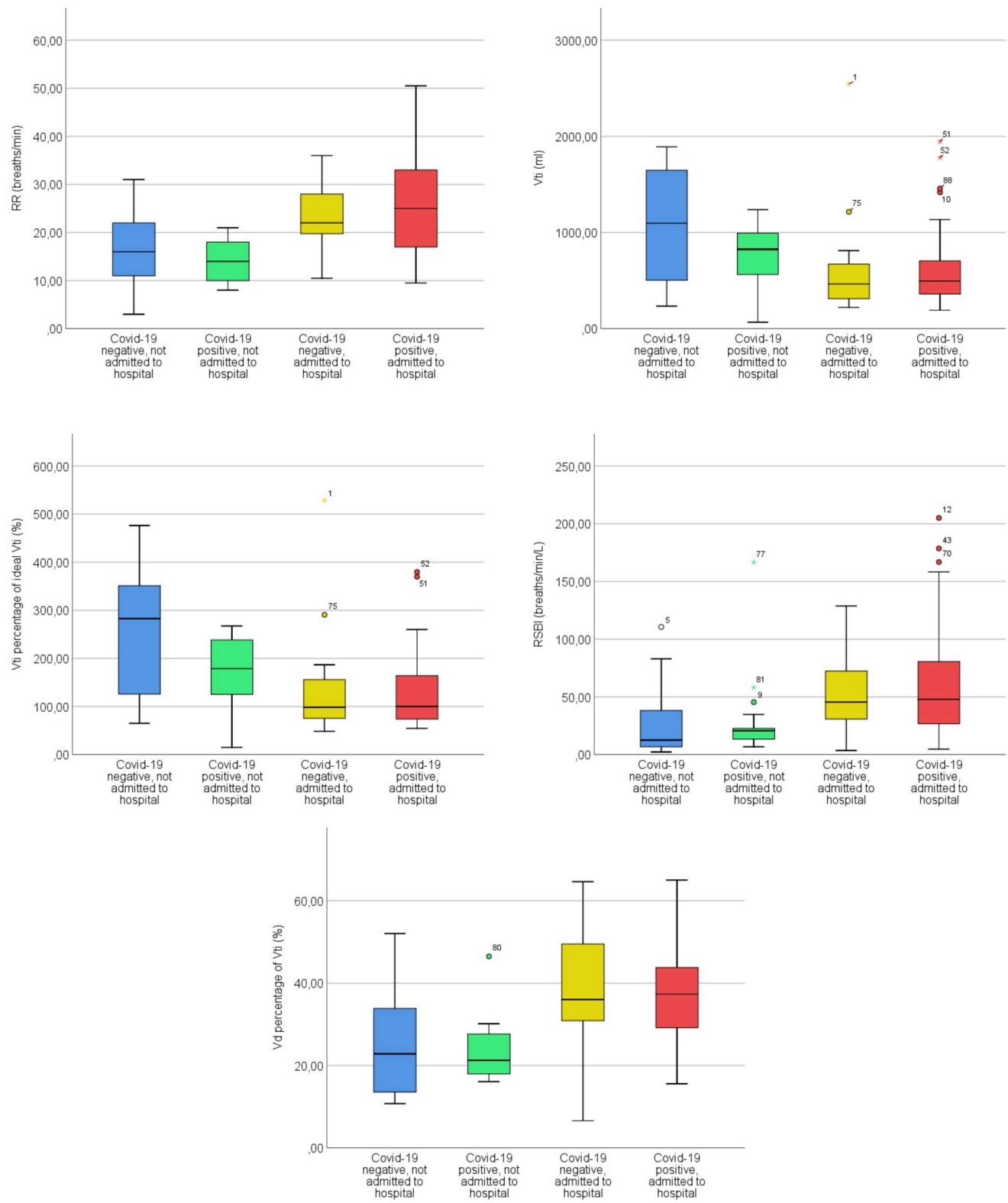

**Fig 2. Boxplots of significant differences in physiological respiratory parameters, hospital admissions and COVID-19.**

**Table 6. Logistic regression models of physiological respiratory parameters effects on the odds of hospital admissions.**

| | Unadjusted model | | Adjusted model[a] | |
|---|---|---|---|---|
| Patient characteristics | Or 95% CI | p-value | OR 95% CI | p-value |
| Age | 1.055 (1.027–1.084) | 0.000* | | |
| Sex | 3.130 (1.314–7.459) | 0.010* | | |
| BMI | 1.004 (0.940–1.071) | 0.916 | | |
| Risk factors categorical | 2.404 (1.342–4.305) | 0.003* | | |
| COVID-19 | 1.749 (0.757–4.043) | 0.191 | | |
| Physiological respiratory parameters | | | | |
| RR (breaths/min) | 1.193 (1.100–1.293) | 0.000* | 1.196 (1.081–1.324) | 0.001* |
| MV (L/min) | 0.997 (0.940–1.056) | 0.907 | 1.050 (0.963–1.145) | 0.268 |
| RSBI (breaths/min/L) | 1.026 (1.010–1.043) | 0.002* | 1.023 (1.006–1.040) | 0.009* |
| Vti (ml) | 0.999 (0.998–1.000) | 0.003* | 0.999 (0.998–1.000) | 0.130 |
| Vti percentage of ideal Vti | 0.992 (0.987–0.997) | 0.001* | 0.996 (0.990–1.002) | 0.172 |
| PIF (L/min) | 0.991 (0.971–1.011) | 0.364 | 1.001 (0.974–1.029) | 0.920 |
| PEF (L/min) | 1.001 (0.984–1.017) | 0.935 | 1.005 (0.983–1.027) | 0.657 |
| NIP (cmH2O) | 1.379 (0.816–2.331) | 0.229 | 0.943 (0.461–1.931) | 0.873 |
| PEP (cmH2O) | 1.086 (0.599–1.971) | 0.786 | 1.299 (0.606–2.785) | 0.502 |
| EtCO2 (kPa) | 0.746 (0.483–1.153) | 0.187 | 0.820 (0.471–1.427) | 0.482 |
| Vd (ml) | 0.999 (0.995–1.004) | 0.831 | 1.000 (0.993–1.006) | 0.900 |
| Vd percentage of Vti (%) | 1.104 (1.053–1.157) | 0.000* | 1.091 (1.027–1.159) | 0.005* |

Abbreviations: IQR: Interquartile range; OR: Odds ratio; CI: Confidence interval; BMI: Body mass index; RR: Respiratory rate; MV: Minute ventilation; RSBI: Rapid shallow breathing index; Vti: Inspiratory tidal volume; PIF: Peak inspiratory flow; PEF: Peak expiratory flow; NIP: Negative inspiratory pressure; PEP: Peak expiratory pressure; EtCO2: End tidal carbon dioxide; Vd: Dead space volume.

[a] Adjusted for age, sex, BMI, risk factors categorical and COVID-19.

* P-value of $\leq$ 0.05.

confirmed cases during the pandemic is 40 years old, which could perhaps in part explain our findings [19]. Another possible explanation is that older patients with COVID-19 treated by the EMS were more seriously ill and were not included in the study as treating them quickly was prioritized. Younger patients with less severe COVID-19 might therefore have been included to a higher degree. This remains speculative though.

Somewhat surprisingly, the oxygen saturation levels did not differ between the patients with and without COVID-19 in this study. Previous pre-hospital studies [4,20] have found lower oxygen saturation levels in all pre-hospital patients with COVID-19 confirmed later on. In our study we had the aim of including all COVID-19 patients and therefore we might have two archetypes of COVID-19 patients in the present study. Firstly the ones with mild to moderate severity of the disease where it is possible that the oxygen saturation levels does not differ compared to other similar patient categories. Secondly the ones with severe COVID-19 where oxygen saturation level might be a stronger predictor. We might have failed to include a sufficient amount of the latter category of patients due to the fact that during the inclusion process the patients with severe COVID-19 were not included due to their inability to leave consent to participate in the study. This however remains speculative.

When looking at COVID-19 patients admitted to hospital and comparing them with COVID-19 patients not admitted the hospitalized population had higher respiratory rate (25 breaths/minute vs. 14 breaths/minute, p<0.001), higher rapid shallow breathing index (47.9 breaths/min/L vs. 20.8 breaths/min/L, p<0.001), lower tidal volume (495.5 ml vs. 826.0 ml, p = 0.010), lower tidal volume percentage of ideal tidal volume (100.2% vs. 178.6%, p = 0.001)

and higher dead space percentage of tidal volume (37.4% vs. 21.3% p<0.001). The odds of hospital admission increased with increasing respiratory rate (OR 1.196, 95% CI 1.081–1.324, p = 0.001), rapid shallow breathing index (OR 1.023, 95% CI 1.006–1.040, p = 0.009) and dead space percentage of tidal volume (OR 1.091, 95% CI 1.027–1.159 p = 0.005). A higher respiratory rate among hospitalized patients seems logical as there is evidence that increased respiratory rate is a predictor for hospital admissions, ICU admissions and mortality [20–23]. Higher dead space fraction has also been found in severe COVID-19 [24] and non COVID-19 related ARDS [25]. These studies were however performed on mechanically ventilated patients and might not be comparable to our findings. Higher dead space fraction in patients with severe COVID-19 has been hypothesized to be linked to pulmonary microvascular endothelial damage and microthrombotic processes causing increases in alveolar dead space [26]. As our study found that patients admitted to hospital with COVID-19 had a high dead space fraction, our results could indicate that the endothelial damage and micro thrombotic processes are present and detectable in the early stages of the disease. Dead space fraction in patients in the pre-hospital setting is an interesting parameter that warrants more research.

To our knowledge this is the first study elucidating the physiological respiratory parameters of patients with COVID-19 in the prehospital setting. Earlier studies have focused on respiratory rate but our goal was to see if physiological respiratory parameters could add valuable information about these patients and potentially aid EMS personnel in the decision of whether to transport a patient to the hospital or leave them at home. Our findings suggest that low tidal volume is a predictor of COVID-19. Increased respiratory rate may however be a more important predictor due to the fact that it seems to be a predictor for admission to hospital. The rapid shallow breathing index might however also be an interesting parameter to follow in these patients as it combines both respiratory rate and tidal volume and remained significant in all our descriptive analyses and logistic regression models.

In this study we could not find clear evidence that silent hypoxemia was always present in patients with COVID-19. Patients with COVID-19 who were admitted to hospital had considerably higher respiratory rate than the ones who were not (25 breaths per minute vs. 14 breaths per minute, p-value < 0.001) which is inconsistent with the presence of silent hypoxemia in this group. A respiratory rate of 25 breaths/minute should cause the patient to feel dyspneic and be easily detectable by an EMS professional. However, levels of EtCO2 did not differ significantly between these groups. Unchanged EtCO2 levels despite low oxygen saturation has been suggested as a contributing mechanism behind silent hypoxia [11], something that our results might be in line with. Of the 53 patients who had COVID-19, 10 of them did not have dyspnea. This implies that silent hypoxemia cannot completely be ruled out in these patients, although they were too few to analyze with any statistical certainty.

We failed to detect any predictors for ICU-admission, possibly due to the limited number of patients admitted to the ICU (8 patients). There were also several other limitations to this study. Since breathing is a controllable process, it cannot be ruled out that some of the patients who performed the sampling procedure changed their way of breathing during the test. A longer sampling time might have adjusted for this but was deemed impractical in this setting. The patient cohorts were also quite small, and larger cohorts would have produced more reliable findings. Due to financial and practical reasons, larger a cohort was not feasible and a goal of 100 included patients was set at the start of the study. Since the EMS personnel themselves performed the inclusion of patients on the basis of COVID-19 symptoms, there is a risk that there was some form of selection bias. As there was no true randomization, the patients included might not be a fully accurate representation of the complete COVID-19 patient cohort that EMS encounter.

The present results indicate that developing a tool that could assist the EMS personnel in the detection of COVID-19 and in the triage of these patients might be beneficial. A device capable of real time measurements of physiological respiratory parameters could be used in this application to evaluate the risk of COVID-19 and hospital admissions. With additional training both in the handling of the device and understanding of the physiological respiratory parameters, the EMS personnel could add the device as an additional tool in the triage of patients with suspected COVID-19.

This study shows that it is possible to measure physiological respiratory parameters in the pre-hospital setting and it indicates that the method has the potential to be used when seeing patients with other diseases of the lung. COPD and asthma are both common diseases encountered by the EMS personnel and a better way to predict the severity of these diseases might increase the correct level of care and by that increase the optimal utility of hospital resources. More research in this field is needed and could present interesting possibilities for emergency medical systems worldwide.

## Conclusion

The patients in this study who took smaller, faster breaths with less pressure and lower flow had higher odds of having COVID-19. Patients admitted to hospital took smaller, faster breaths than those who were not admitted, and increased respiratory rate, rapid shallow breathing index and dead space percentage of tidal volume increased the odds of hospital admission. This study presents a novel way of examining patients with suspected COVID-19 in the prehospital setting, and the method could potentially be used as a triage tool for these patients. More research is needed to confirm these findings and to investigate if this method is feasible in other diseases of the lung, such as other types of pneumonia, COPD and asthma.

## Supporting information

**S1 Dataset. Full statistical analysis data for Table 1.**
(XLSX)

**S2 Dataset. Full statistical analysis data for Table 2.**
(XLSX)

**S3 Dataset. Full statistical analysis data for Table 3.**
(XLSX)

**S4 Dataset. Full statistical analysis data for Table 4.**
(XLSX)

**S5 Dataset. Full statistical analysis data for Table 5.**
(XLSX)

**S6 Dataset. Full statistical analysis data for Table 6.**
(XLSX)

**S7 Dataset. Minimal data set.**
(XLSX)

## Acknowledgments

The authors would like to thank the personnel of Uppsala emergency medical services for including patients in this study during the trying times of the pandemic.

## Author Contributions

**Conceptualization:** Johan Mälberg, David Smekal.

**Data curation:** Johan Mälberg, Nermin Hadziosmanovic.

**Formal analysis:** Johan Mälberg, David Smekal.

**Investigation:** Johan Mälberg, David Smekal.

**Methodology:** Johan Mälberg, Nermin Hadziosmanovic, David Smekal.

**Project administration:** David Smekal.

**Resources:** Johan Mälberg, David Smekal.

**Software:** Nermin Hadziosmanovic.

**Supervision:** David Smekal.

**Validation:** Johan Mälberg, Nermin Hadziosmanovic.

**Visualization:** Johan Mälberg.

**Writing – original draft:** Johan Mälberg, David Smekal.

**Writing – review & editing:** Johan Mälberg, Nermin Hadziosmanovic, David Smekal.

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
