## [Decision Letter · Decision Letter 0]

26 Jul 2021

PONE-D-21-15446

Respiratory mechanics in pre-hospital patients with suspected covid-19: A prospective cohort study

PLOS ONE

Dear Dr. Mälberg,

Thank you for submitting your manuscript to PLOS ONE. After careful consideration, we feel that it has merit but does not fully meet PLOS ONE’s publication criteria as it currently stands. Therefore, we invite you to submit a revised version of the manuscript that addresses the points raised during the review process.

This a well conducted investigation, dealing with a hot topic. We suggest the Authors to better discuss the clinical significance of capnography in this specific clinical setting. A paragraph describing the potential clinical impact of the present results should be added. 

We look forward to receiving your revised manuscript.

Kind regards,

Chiara Lazzeri

Academic Editor

PLOS ONE

Journal Requirements:

2. During your revisions, please note that a simple title correction is required: please correct "covid-19" to "COVID-19". Please ensure this is updated in the manuscript file and the online submission information.

4. We note that Figure 1 in your submission contain copyrighted images. All PLOS content is published under the Creative Commons Attribution License (CC BY 4.0), which means that the manuscript, images, and Supporting Information files will be freely available online, and any third party is permitted to access, download, copy, distribute, and use these materials in any way, even commercially, with proper attribution. For more information, see our copyright guidelines: http://journals.plos.org/plosone/s/licenses-and-copyright.

Reviewers' comments:

Reviewer's Responses to Questions

**Comments to the Author**

1. Is the manuscript technically sound, and do the data support the conclusions?

Reviewer #1: Yes

2. Has the statistical analysis been performed appropriately and rigorously? 

Reviewer #1: Yes

3. Have the authors made all data underlying the findings in their manuscript fully available?

Reviewer #1: Yes

4. Is the manuscript presented in an intelligible fashion and written in standard English?

Reviewer #1: Yes

5. Review Comments to the Author

Reviewer #1: The authors present the results of a prospective cohort study in the pre-hospital setting focusing on physiopathological respiratory parameters in patients with suspected COVID-19. They mainly found an association between COVID-19 diagnosis and/or need for hospitalization from one part and rapid shallow breathing and negative inspiratory pressure from another part. They also report an association between physiological dead-space and hospital admission. The authors should be congratulated for such a difficult to achieve clinical study. I have some comments aiming to clarify some issues and to help do discuss some results.

Major comments

1. The authors should be congratulated for the study, permitting to appreciate in part both gas exchanges and volume/pressure parameters. In this way, since severity of COVID-19 is predominantly linked to respiratory failure, it could be interesting to display and discuss more deeply oxygen saturation results, not only for description of the included population but also for appreciating the respective interests of SaO2, capnography and spirometry results for prediction of COVID-19 diagnosis and severity.

2. The capnography results (link to COVID-19 severity) could be discussed in light of the hypothesis of large amounts of alveolar dead-space in relation to lung microvascular endothelialitis and microthrombosis. These aspects are discussed in the following reference: Respiratory mechanics and gas exchanges in the early course of COVID-19 ARDS: a hypothesis-generating study. Ann Intensive Care. 2020 Jul 16;10(1):95. doi: 10.1186/s13613-020-00716-1. More generally, there are various arguments for such a hypothesis (see as an example: COVID-19 is a systemic vascular hemopathy: insight for mechanistic and clinical aspects. Angiogenesis. 2021 Jun 28:1-34. doi: 10.1007/s10456-021-09805-6.).

Minor comments

1. The tittle could be modified to “Physiological respiratory parameters in pre-hospital patients: …”. Indeed, respiratory mechanics typically do not include physiological dead-space measurements and is best appreciated by both pressure and volume measurements, including in spontaneously breathing patients esophageal pressure measurements. The same apply for Table legends, etc.

2. The interest of Figure 1 seems to me debatable.

3. Ref. 14 refers to a 6ml/kg PBW, not 7.

4. Ref 25 doesn’t refer to a COVID-19 study.

6. PLOS authors have the option to publish the peer review history of their article (what does this mean?). If published, this will include your full peer review and any attached files.

Reviewer #1: No

---

## [Author Response · Author response to Decision Letter 0]

17 Aug 2021

Response to reviewers

The authors would like to thank the editor and reviewer for their hard work and valuable suggestions for improvement regarding our manuscript. Please see the answers to each individual comment below. We hope that you will find the revised manuscript suitable for publication in PLOS ONE.

Sincerely

Johan Mälberg

Editor’s comments:

This a well conducted investigation, dealing with a hot topic. We suggest the Authors to better discuss the clinical significance of capnography in this specific clinical setting. A paragraph describing the potential clinical impact of the present results should be added

Author response:

We have discussed the clinical significance of capnograhy in greater detail, line 279-285.

We have added two paragraphs discussing the potential clinical impact of the study’s findings, line 318-329.

Journal Requirements:

Author response:

We have updated the manuscript in accordance with the submission guidelines and style requirements. File names have been revised in accordance to the guidelines. Please see the revised manuscript with tracked changes for all individual changes.

2. During your revisions, please note that a simple title correction is required: please correct "covid-19" to "COVID-19". Please ensure this is updated in the manuscript file and the online submission information.

Author response:

We have corrected covid-19 to COVID-19.

Author response:

We have updated the cover letter with information regarding the limits in our ethical approval that prohibits us from sharing our full minimal data set. In order to share as much of our underlying data as possible, we have blocked certain information in our minimal data set to prevent individual participants from being identifiable. This was done after consulting the Swedish Ethical Review Authority in order comply with our ethical approval, but also with Swedish law and the EU’s general Data Protection Regulation.

We have added the minimal data set as supporting information and updated our data availability section to reflect this.

The previous data availability statement was due to a misunderstanding on the corresponding authors’ part. We apologize for the mistake.

4. We note that Figure 1 in your submission contain copyrighted images. All PLOS content is published under the Creative Commons Attribution License (CC BY 4.0), which means that the manuscript, images, and Supporting Information files will be freely available online, and any third party is permitted to access, download, copy, distribute, and use these materials in any way, even commercially, with proper attribution. For more information, see our copyright guidelines: http://journals.plos.org/plosone/s/licenses-and-copyright.

Author response:

Figure 1 was a picture taken by the corresponding author. It has been removed in accordance with the suggestion from reviewer 1. The remaining figures have been renamed accordingly.

Review comments to authors

Reviewer #1: The authors present the results of a prospective cohort study in the pre-hospital setting focusing on physiopathological respiratory parameters in patients with suspected COVID-19. They mainly found an association between COVID-19 diagnosis and/or need for hospitalization from one part and rapid shallow breathing and negative inspiratory pressure from another part. They also report an association between physiological dead-space and hospital admission. The authors should be congratulated for such a difficult to achieve clinical study. I have some comments aiming to clarify some issues and to help do discuss some results.

Author response:

We would like to thank the reviewer for the kind words and the valuable comments.

Major comments

1. The authors should be congratulated for the study, permitting to appreciate in part both gas exchanges and volume/pressure parameters. In this way, since severity of COVID-19 is predominantly linked to respiratory failure, it could be interesting to display and discuss more deeply oxygen saturation results, not only for description of the included population but also for appreciating the respective interests of SaO2, capnography and spirometry results for prediction of COVID-19 diagnosis and severity.

Author response:

We have added a discussion concerning our SaO2 findings, line 256-266. We agree that SaO2 is an interesting parameter with regards to COVID-19 even though we did not find any significant differences with regard to the SaO2 levels in our study.

2. The capnography results (link to COVID-19 severity) could be discussed in light of the hypothesis of large amounts of alveolar dead-space in relation to lung microvascular endothelialitis and microthrombosis. These aspects are discussed in the following reference: Respiratory mechanics and gas exchanges in the early course of COVID-19 ARDS: a hypothesis-generating study. Ann Intensive Care. 2020 Jul 16;10(1):95. doi: 10.1186/s13613-020-00716-1. More generally, there are various arguments for such a hypothesis (see as an example: COVID-19 is a systemic vascular hemopathy: insight for mechanistic and clinical aspects. Angiogenesis. 2021 Jun 28:1-34. doi: 10.1007/s10456-021-09805-6.).

Author response:

We have added a discussion of the capnography results and dead space and added the first mentioned article as a reference, line 279-285. We find this study’s hypotheses very interesting in combination with our findings, which could implicate that the damage to the lungs is detectable early on with our method of measuring the physiological respiratory parameters.

Minor comments

1. The tittle could be modified to “Physiological respiratory parameters in pre-hospital patients: …”. Indeed, respiratory mechanics typically do not include physiological dead-space measurements and is best appreciated by both pressure and volume measurements, including in spontaneously breathing patients esophageal pressure measurements. The same apply for Table legends, etc.

Author response:

We agree with the reviewer and have changed the title and the term used in the manuscript to physiological respiratory parameters.

2. The interest of Figure 1 seems to me debatable.

Author response:

We agree and have removed the figure.

3. Ref. 14 refers to a 6ml/kg PBW, not 7.

Author response:

The reviewer is correct. Most sources cite the normal value of tidal ventilation in healthy adults as 6-8 ml/kg. This should have been made clearer. We have changed reference 14 to one that in detail describes the basis of 7 ml/kg and how it came to be.

4. Ref 25 doesn’t refer to a COVID-19 study.

Author response:

We agree that the way the references were mentioned it was not clear that reference 25 did not concern COVID-19. We have changed this to make it clearer. Even though the article does not concern COVID-19, we think it adds value as it shows that changes in dead space occur in ARDS, which severe COVID-19 can cause

---

## [Editor Report · Decision Letter 1]

23 Aug 2021

Physiological respiratory parameters in pre-hospital patients with suspected COVID-19: A prospective cohort study

PONE-D-21-15446R1

Dear Dr. Mälberg,

We’re pleased to inform you that your manuscript has been judged scientifically suitable for publication and will be formally accepted for publication once it meets all outstanding technical requirements.

Kind regards,

Chiara Lazzeri

Academic Editor

PLOS ONE
---

## [Editor Report · Acceptance letter]

26 Aug 2021

PONE-D-21-15446R1 

Physiological respiratory parameters in pre-hospital patients with suspected COVID-19: A prospective cohort study 

Dear Dr. Mälberg:

I'm pleased to inform you that your manuscript has been deemed suitable for publication in PLOS ONE. Congratulations! Your manuscript is now with our production department. 

Kind regards, 

on behalf of

Dr. Chiara Lazzeri 

Academic Editor

PLOS ONE